# Automatic Alignment Method for Controlled Free-Space Excitation of Whispering-Gallery Resonances

**DOI:** 10.3390/s23219007

**Published:** 2023-11-06

**Authors:** Davide D’Ambrosio, Marialuisa Capezzuto, Antonio Giorgini, Pietro Malara, Saverio Avino, Gianluca Gagliardi

**Affiliations:** Consiglio Nazionale delle Ricerche, Istituto Nazionale di Ottica (INO), Via Campi Flegrei, 34 Comprensorio A. Olivetti, I-80078 Pozzuoli, Italy; marialuisa.capezzuto@ino.cnr.it (M.C.); antonio.giorgini@ino.cnr.it (A.G.); pietro.malara@ino.cnr.it (P.M.); saverio.avino@ino.cnr.it (S.A.); gianluca.gagliardi@ino.cnr.it (G.G.)

**Keywords:** whispering-gallery mode resonator, free-space coupling, Mie scattering, automatic alignment, spatial light modulator

## Abstract

Whispering-gallery mode microresonators have gained wide popularity as experimental platforms for different applications, ranging from biosensing to nonlinear optics. Typically, the resonant modes of dielectric microresonators are stimulated via evanescent wave coupling, facilitated using tapered optical fibers or coupling prisms. However, this method poses serious shortcomings due to fabrication and access-related limitations, which could be elegantly overcome by implementing a free-space coupling approach; although additional alignment procedures are needed in this case. To address this issue, we have developed a new algorithm to excite the microresonator automatically. Here, we show the working mechanism and the preliminary results of our experimental method applied to a home-made silica microsphere, using a visible laser beam with a spatial light modulator and a software control.

## 1. Introduction

Liquid and solid whispering-gallery mode microresonators (WGMRs) have emerged during recent decades as key optical elements involved in a plethora of applications, from sensing to non-linear optics and optomechanics [1,2]. The vast majority of these experiments are based on evanescent wave excitation schemes; they rely on devices such as tapered fibers or prisms, where frustrated total internal reflection permits highly efficient light coupling to WGMRs [3,4]. This approach proves to be ideally suited for the class of experiments requiring a high circulating power [5,6,7,8]. Nonetheless, these coupling devices suffer from several drawbacks, such as their limited compatibility with liquid resonators and their lack of accessibility in general, as well as the lack of robustness and the short lifetime of the fiber taper-based versions.

If the requirement of efficient coupling is not crucial, free-space WGM excitation can make the experimental setup more robust, reliable, and versatile. In particular, in experiments involving delicate liquid microresonators [9], the absence of external coupling devices in close proximity represents a key advantage. Free-space coupling relies on angular momentum matching between a Gaussian radiation beam and the whispering-gallery mode (WGM) resonances of a dielectric sphere, as shown in [9]. Based on the Lorenz–Mie scattering theory, the interaction regime depends upon the ratio between the sphere radius (r) and the displacement of the laser beam (d) from the center of the sphere. It is possible to excite almost pure WGMs by focusing a laser beam tangentially to the surface of the sphere, with r/d ≈ 0.95 (i.e., off-axis excitation). However, the manual ability required for such alignment is non-negligible and, in principle, could limit the applicability and repeatability of free-space schemes. For instance, a recent unpublished work has proposed to optimize backscatter collection using a digital micromirror device for the free-space excitation of silica microtoroids [10]. Here, we present a novel automatic alignment method that enables optimal WGM free-space coupling to a microresonator. A spatial light modulator (SLM) allows us to tilt the direction of the input beam, setting an injection condition; a charge-coupled device (CCD) collects and analyses the light laterally scattered via the illuminated microsphere; and a custom-built algorithm evaluates the WGM coupling. The iteration of such a process generates a hierarchic map of the alignment conditions, which is used by the algorithm to select the highest-Q mode available. The robustness and reproducibility of the automatic alignment are ensured for N > 100 loop iterations.

## 2. Experimental Setup

A distributed feedback (DFB) diode laser operating at 640 nm impinges on a phase-only SLM (Thorlabs Exulus HDD2) and is reflected with an angle r. The laser beam is then focused using a 10× microscope objective on the edge of a home-made silica microcavity. The cavity consists of a spherical bulb fabricated on the tip of a single-mode optical fiber (outer diameter 125 μm). Its end is arc-fused using the electrodes of a fiber splicer. The melting of the silica creates an almost spherical bulge, with a radius ranging from a few tens of micrometers to hundreds of micrometers. The forward-transmitted and backscattered beams are collected and detected via two fast photodiodes, while the light laterally scattered from the microsphere is imaged through a 1:1 telescope on a CCD camera (Basler mod. a2A1920-160ucBAS, 1920 × 1200 pixels). Additional details regarding the imaging system can be found in [11]. A fast frequency scan of the laser frequency allows us to visualize the spectrum of the excited WGMs through the photodiode signal. Instead, the CCD image, having a much longer integration time, displays the total number of photons scattered from all the modes of the spectrum instantaneously. Typically, scattering patterns from whispering gallery resonances consist of light rings strongly confined along the microresonator’s circumference. 

The core of the automatic alignment system is a custom-built algorithm that analyses the scattering pattern of the CCD and generates a phase mask signal for the SLM to change the injection angle r (an example of such a phase mask is shown in Figure 1). The algorithm assigns a figure of merit to each scattering pattern (i.e., to each alignment condition) by summing up the pixel values within a ring-shaped region of interest (ROI). The ROI filter is applied to only select the light scattered from the WGM region, thus removing spurious contributions to the light intensity measurement. The calculated total value is indeed proportional to the total intensity scattered from the microresonator circumference, i.e., from the injected WGM mode. This figure of merit was chosen for its robustness against spurious signals. As shown in the example of Figure 2b, the alignment optimization converges successfully, even in the presence of the two clearly visible spots caused by stray reflections on the focusing and collection objectives. 

For each value of the parameter r generated from a given a SLM phase mask, the algorithm calculates the figure of the merit value from the CCD data, stores it (together with the phase mask), and generates a new alignment. After about 100 of these iterations, a reliable map of alignments, ranked according to their level of WGM-coupled intensity, is available to the algorithm, which can then automatically select the brightest mode.

An alternative way to carry out an automatic alignment could be based on a direct feedback routine, where at each step, the tilt correction of the SLM is proportional to the detected increment/decrement of the figure of merit. However, an inherent drawback of these iterative procedures, although they are widely adopted for a variety of optimization tasks, is their strong dependence upon the initial conditions. In our WGM alignment task, a non-ideal estimate of the initial parameters would either lead to divergence or bring the system to low-Q modes, corresponding to one of the many relative maxima of the merit functions. Thus, here we followed an alternative approach which prevents this issue by implementing a random guess of the tilt parameter at each step. In addition to being independent of the initial conditions, we note that with this random walk approach, the system only requires a reasonable amount of time to consistently find the brightest mode, as discussed in the next section. 

## 3. Results

In Figure 3, we report some of the obtained spectra. The plots are normalized for ease of visualization, but their signal to noise ratio clearly shows that if the loop iteration number (N) is high enough, the system tends to always align on the brightest mode, with a Q-factor of 2.3 × 10^6^ (third panel, from top to bottom). Instead, for lower values of N, the system may occasionally converge to some lower-Q modes. The WGMs shown in panels 1 and 2, from the top, exhibit Q-factors of 1.5 and 2.0 × 10^6^, respectively. The algorithm appears to be robust and highly reproducible for N values higher than 100. The nominal frame rate of our SLM is 60 Hz, while the CCD frame rate is 50 Hz. Taking into account the delay caused by the communication bus with the computer and computer processing time, the alignment procedure ends up with a maximum frequency of about 2 Hz. Considering this, and the threshold of N = 100, we assess that our method can automatically align a microresonator on a high-Q whispering-gallery mode in ≈60 s. We point out that the system shows a resolution of ≈0.3 mrad, which is the off-axis deviation corresponding to a single line increment in the phase mask grating. Also, the effective SLM scan range of the angle r is limited by the maximum permissible deflection within the focusing optics in front of the microresonator, and corresponds to about 15 mrad.

Finally, in Figure 4, we report the resonant spectra of a different microresonator, which shows split modes; the splitting is likely due to the presence of impurities on the surface of the silica microsphere, which act as scatterers. Even if dust particles happen to be in the CCD’s ROI, the alignment algorithm remains just as robust. In this condition, the laser beam is no longer able to excite a single high-Q equatorial mode [12], but we find the same dependency of the alignment reproducibility upon N as above.

## 4. Conclusions

The measurements presented in this work demonstrate that a compact optical setup, comprising an SLM and a CCD camera, can be implemented to automatically align a whispering-gallery mode microresonator with high repeatability and robustness. The proposed scheme relies on a simple one-parameter process, but its performance can be readily improved by increasing the dimensions of the parameter space. For example, a second optimization step could be adjusting the laser tilt angle in the orthogonal direction. To obtain this second degree of freedom, a phase mask identical to the one shown in Figure 1, but tilted by 90 deg, can be overlapped onto the SLM. A third, parallel alignment cycle could optimize the focusing of the beam on the edge of the resonator by introducing an additional phase mask acting as a Fresnel lens and randomly scanning its focal length. Also, from a careful analysis of the communication bus-induced delays, it is concluded that using dedicated hardware, such as a Field-Programmable Gate Array (FPGA), would significantly enhance the overall frame rate, bringing the setup close to the ultimate limit given by the CCD and SLM refresh rates (~50 Hz). 

Whatever the degree of complexity chosen, the described procedure can be completely automated, which provides great benefits for applications where the main goal is to keep the system in operation for long times even in the presence of external perturbations. As an example, in refs [13,14], the laser frequency is tuned to lay on the side of a cavity mode. Any shifts of the excited resonance due to proteins or molecules binding is often retrieved by detecting the corresponding variation of the transmitted light intensity. In this case, it is crucial for the measurement of performance to rule out any false signal due to intensity drifts, as well as reducing amplitude noise caused by coupling fluctuations, which requires keeping the laser alignment to the whispering-gallery resonator as stable as possible. This automated alignment of microcavities, eventually combined with externally controlled or structured light beams, envisages a new perspective for exploiting the selective excitation of whispering-gallery modes via a novel information-encoding process with optical methods.

## Figures and Tables

**Figure 1 sensors-23-09007-f001:**
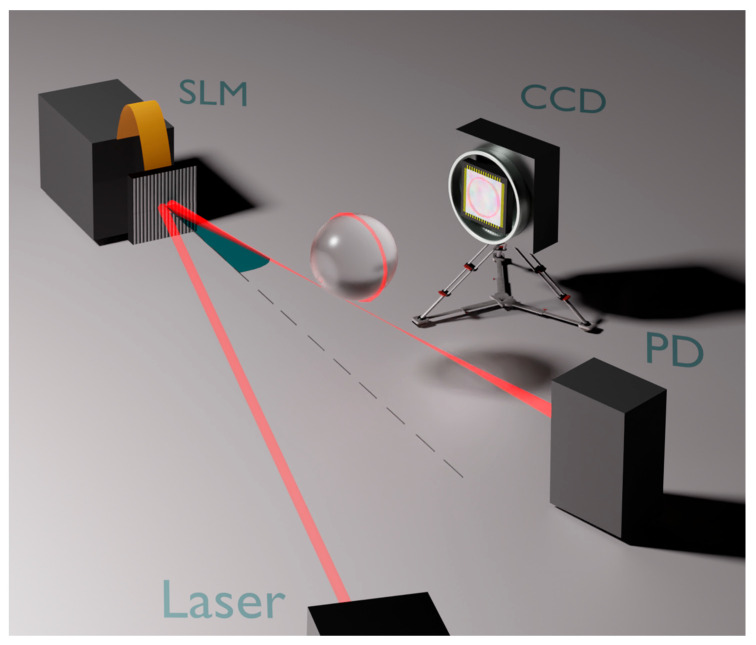
Experimental setup. A grating-like phase mask is imaged onto the SLM panel to tilt the laser beam by acting on the reflection angle r (shown in turquoise). Light scattered from the resonator is imaged using a CCD camera through a 1:1 telescope. The telescope, and a second photodiode (PD) for backscattering detection, are omitted for sake of clarity.

**Figure 2 sensors-23-09007-f002:**
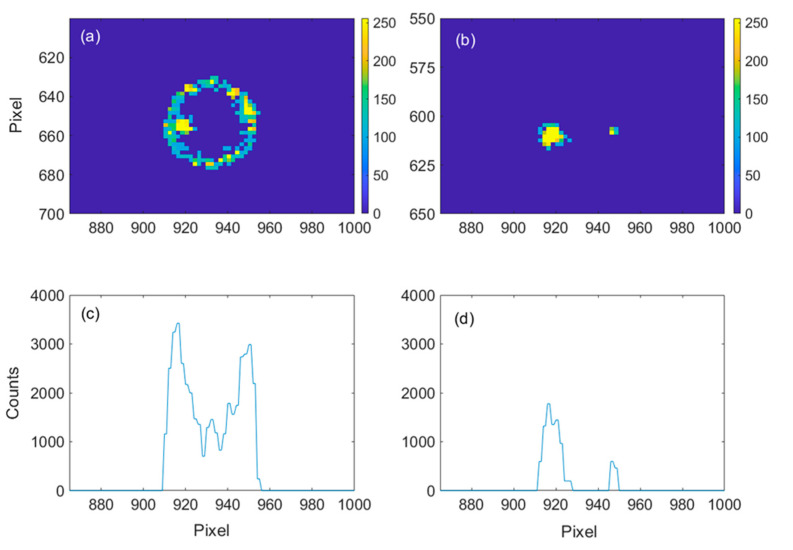
Images recorded with the CCD camera when a WGM is excited (**a**) or not (**b**). On the corresponding lower panels (**c**,**d**), we report a 1 D integration of the total number of counts. We ruled out via software the contribution of the two stray light spots, visible in both upper panels, so as to obtain a robust selection criterion to distinguish among the system’s available alignment conditions.

**Figure 3 sensors-23-09007-f003:**
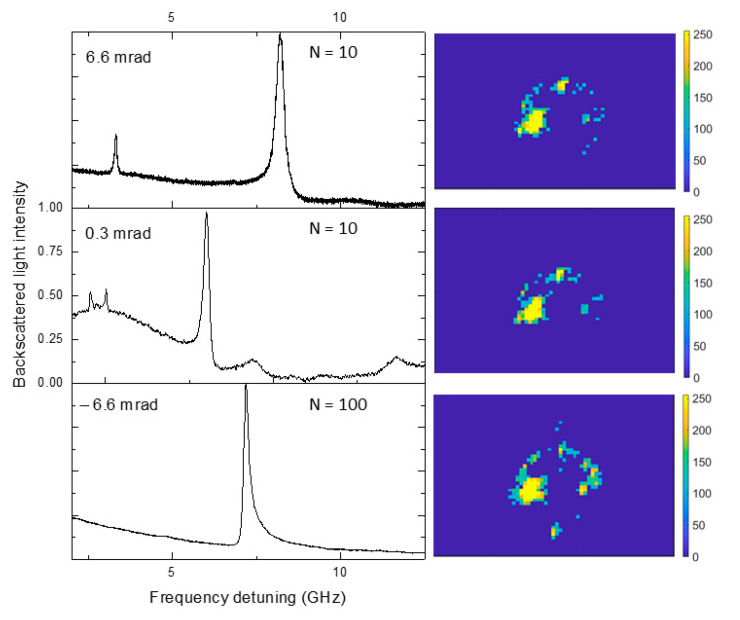
Microresonator’s self-aligned modes, normalized for ease of visualization. In each plot, the tilt angle and the number of iterations are also reported. For N = 100 iterations of the algorithm, the system tends to align on the mode in the lower panel, while for lower N values, the system finds local maxima of the scattered light that correspond to lower-Q whispering-gallery modes (from the top, panels 1 and 2).

**Figure 4 sensors-23-09007-f004:**
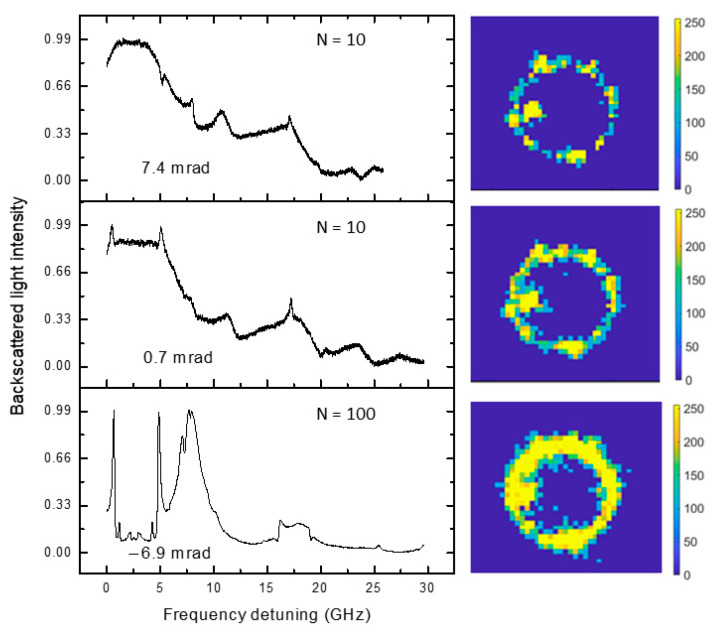
Microresonator’s self-aligned modes in the presence of dust particles (normalized). Split modes are visible in the spectra, revealing the presence of scatterers on the surface of the WGMR.

## Data Availability

The data presented in this study are available on request.

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
