# Peer review of "Automatic Alignment Method for Controlled Free-Space Excitation of Whispering-Gallery Resonances"

_sensors, 2023, doi:10.3390/s23219007_

Round 1
Reviewer 1 Report
Comments and Suggestions for Authors
D’Ambrosio et al, present work on free-space coupling using microsphere resonators. The work appears technically correct; however, the basis for this work is essentially presented in an arXiv article that was published on August 1 on using a DMD to free-space couple light into a microtoroid resonator.
I recommend publication pending citation and appropriate attribution of this work: https://arxiv.org/abs/2308.00726
In addition, this paper, which was also published on Sensors should be cited:
https://www.mdpi.com/1424-8220/23/13/5925
It is on free-space coupling into microsphere resonators.
The Q-values that are obtained should be mentioned as well.
Reviewer 2 Report
Comments and Suggestions for Authors
The paper describes an alignment method for an optical whispering gallery micro-resonator, based on randomized displacement of the exciting laser beam and automated image processing of the stray light.
To the best of my knowledge, the work is novel and original. I think that the title of the paper is appropriate. In the abstract, the work is motivated, the approach is summarized and the main results are mentioned. In the introduction, the necessity and relevance of the work is appropriately motivated and it is put into context of prior publications of others and of the authors. I think that in section 2 on experimental setup, a lot of important information is missing, see my comments below. Particularly, more details on components, on the geometry and on the image processing and the automated algorithm need to be given. In section 3 on results, I missed some detailed explanations of some of the measurement results. In the conclusion, the results are briefly summarized and an outlook on possible future extension is given.
In revision, the following points should be addressed:
1) Lines 55-56: “home-made silica microcavity, a spherical bulb (~100-µm diam) fabricated on the tip of an optical fiber by a fusion splicer”. From this sentence, I did not understand how the microcavity was fabricated. Please explain in detail how it is done, or provide a reference to a paper where it is explained in detail.
2) Lines 59-60: “Additional detail can be found in [11].” I carefully read [11] but I could not find additional details. In particular, I missed the adjustable range of angles r that the SLM can cover, and the distance between SLM and sphere, or the achievable range of displacements d at the sphere. Reference [11] does not give this information either, and it describes a silicon oil droplet as a WGM resonator, not a silica sphere, plus the geometry of the setup seems to be different, when comparing Figure 5 of [11] with Figure 1 in the present paper.
3) Figure 1: I assume the turquoise circle slice is supposed to illustrate the angle r. Please add an “r” in the illustration to clarify that. From the Figure, it looks like the SLM changes the horizontal angle, however for changing the displacement d of the laser beam, the vertical angle of the laser beam should be changed if the WGM mode is observed in horizontal direction. In [11], the mirror is used to change the direction. Please clarify.
4) Line 76: you mention a “ring-shaped Region Of Interest (ROI)”, however no explanation is given how this ROI looks in detail. Please explain. In particular, please specify its inner and outer radius. How do you calculate the “calculated total value”? Do you simply sum up the intensities of all pixels in that circle ring? Please explain.
5) Figure 2 a) and c): From the text, I understood you evaluate a ring-shaped ROI, see my comment 4, however Figure 2c suggests that you calculated the “Counts” depicted on the vertical scale by simply summing the intensities of the pixels in the vertical columns of Figure 2a. Is that true? If yes, why do you do that? Shouldn’t you use a ring mask, as suggested in line 76 of your manuscript? The same question applies to b) and d), of course.
6) I missed an explanation of how your software works. I just found a few bits and pieces somewhat distributed over page 3. You mention “a random guess of the tilt parameter” (line 95) but don’t give the range out of which the angle (or displacement) is being randomly selected. You mention “We ruled out via software the contribution of the two stray light spots” (lines 100-101) but don’t explain how you ruled them out. Did you exclude the area of these spots from your “ring-shaped ROI”? In line 78, you mention a “figure of merit” but don’t explain it in detail. I don’t understand at all your phrase “generated by given a SLM phasemask” in line 82, which seems to also contribute to your “figure of merit”. A detailed explanation of your algorithm is needed. You should include some information on the image processing you are performing, maybe with the help of an illustration.
7) Lines 110-112: you mention that your SLM and your CCD are fast (50-60 Hz) but communication and signal processing limit your alignment procedure to 2 Hz frame rate. No information is given on which communication bus and which computer hardware and software you use. From looking at the number of pixels (100 x 170) with 8 bit depth, it will be no problem at all to handle the image processing with some dedicated hardware (e.g. a field-programmable gate array). I think it will even be possible to reach 50 Hz per frame with a powerful standard computer and with software optimized for speed. That would shorten the time for the whole alignment procedure with N = 100 to 2 seconds. If this is correct, I suggest that you mention that, for instance in the outlook at the end of the paper.
8) Figure 3, left plots: it should be mentioned, e.g. in the caption, that the horizontal “Frequency” axis denotes the detuning frequency of the Laser. I don’t understand why the resonance peaks show up as positive peaks. I expected negative peaks because the transmitted intensity should be reduced when the WGM mode is excited, as nicely depicted in Figure 7 (left) of Reference [11]. Please explain. In addition, the bottom right image does not really look different from the top right and middle right images. To me, they all look like “lower-Q WG modes”, in contrast to Figure 2 a) which really looks like a nice circular WGM. Please comment on that, too.
9) Figure 4: to me, the three images “in presence of dust particles” look much better than that without dust in Figure 3 (bottom). Can you explain why that is so?
Reviewer 3 Report
Comments and Suggestions for Authors
The manuscript by D'Ambrosio et al. describes an automated iterative convergent method of free space excitation of WGM in a glass microsphere. The paper seems scientifically sound, with the initial hypothesis supported by the experimental observations. It may be of interest to broader applied optics community. My only remark is that the Figure 1 is not completely clear, I would request the authors to mark all components of their setup in the figure, with all essential parameters (like the tilt angle) marked as well. I would also like the authors to comment on the applicability of their iterative alignment method to the measurements requiring fast re-alignment: the authors claim that full re-alignment with ~100 iterations takes up to one minute. Would it rule out some particular applications?
Comments on the Quality of English Language
The quality of the English language is decent, some minor editing would be desirable.
Reviewer 4 Report
Comments and Suggestions for Authors
The Authors report a new algorithm to excite the microresonator automatically. The results have been achieved through experiments by using commercial devices. Althought the manuscript is well written with very interesting results, the Authors have to address the following issues to deserve the manuscript acceptance recommendation:
1. The Authors reports on a microsphere as WGR. However, from the title , the proposed algorithm would seem versatile to all WGRs. Therefore, the Authors has to change the manuscript title and have to report in the introduction also other approaches of WGR excitation (by including RR, wedge resonators and so on). The main trend moves towards the development of integrated platform of excitation WGRs by including waveguides and minimizing the fiber-to-chip losses (see, A broadband chip-scale optical frequency synthesizer at 2.7× 10− 16 relative uncertainty. Science advances, 2(4), e1501489, 2016; Brunetti, G., Heuvink, R., Schreuder, E., Armenise, M. N., & Ciminelli, C. (2023). Silicon Nitride Spot Size Converter with Very Low-Loss over the C-Band. IEEE Photonics Technology Letters, 2023; Low-loss and misalignment-tolerant fiber-to-chip edge coupler based on double-tip inverse tapers. In Optical Fiber Communication Conference (pp. M2I-6). Optica Publishing Group, 2016).
2. The main limitations of the proposed algorithm have to be reported. In particular, which is the maximum Q-factor excitable by the following system?
Round 2
Reviewer 2 Report
Comments and Suggestions for Authors
In revision, the manuscript has been substantially improved. A description of the fabrication of the microcavity has been added. Reference to the prior works [11] has been inproved, pointing out that it refers only to details on the imaging system. A mention of the second photodiode that records the backscattered light has been included. The definition of the region of interest has been detailed further. A sentence of outlook to the possibility of further speeding up the measurement by implementing dedicated hardware has been added.
However, some answers and explanations to my questions that you gave me in your response letter, were not mentioned nor implemented in the revised manuscript. I think it would improve your manuscript if you at least partially included your answers to my comments 5, 6 and 9 in your paper, too, so that not only I, but also all other future readers of your paper profit from your comprehensive explanations.
In addition, I suggest that you address the following points in your revision:
1) Line 65: there is a phrase as “We also collect” which seems to be erroneously placed. Please check and correct that.
2) Caption of Figure 2 in lines 105-106: the sentence “On the corresponding lower panels ((c) and (d)), a 1-D integration of the total number of counts.” is incomplete, a verb is missing.
3) Line 123: I think the word “excursion” is not suitable here, “deflection” would be much better. I suggest to rather write something like: “… scan range … is limited by the maximum permissible deflection within the focusing optics …”.
4) Figures 3 and 4: in your response letter, you comprehensively explain that you use a second photodiode to measure the backscattered light, and you also added that to the manuscript. However, the vertical axis titles in Figures 3 and 4 still read “Normalized transmission”. I think it should be corrected to “Backscattered light intensity”, or similar.
5) Lines 150-152: if the data transfer speed is enhanced, the frame rate is increased, not shortened. I suggest to re-formulate the sentence as: “From a careful analysis of the communication bus-induced delays, it appears that the use of dedicated hardware, such as a Field-Programmable Gate Array (FPGA), would significantly enhance the overall frame rate, bringing …”.
Reviewer 4 Report
Comments and Suggestions for Authors
The Authors have clarified the open and debated points.
Author Response
We thank the reviewer for its comments